# PPP-Net: Platform-aware Progressive Search for Pareto-optimal Neural Architectures

**Jin-Dong Dong**[1], **An-Chieh Cheng**[1], **Da-Cheng Juan**[2], **Wei Wei**[2] **& Min Sun**[1]
National Tsing-Hua University, Hsinchu, Taiwan[1]
Google, Mountain View, CA, USA[2]
`mark840205@gmail.com`
`{dacheng,wewei}@google.com`
`{anjiezheng@gapp,sunmin@ee}.nthu.edu.tw`

## Abstract

Recent breakthroughs in Neural Architectural Search (NAS) have achieved state-of-the-art performances in many applications such as image recognition. However, these techniques typically ignore platform-related constrictions (*e.g.*, inference time and power consumptions) that can be critical for portable devices with limited computing resources. We propose **PPP-Net**: a multi-objective architectural search framework to automatically generate networks that achieve Pareto Optimality. PPP-Net employs a compact search space inspired by operations used in state-of-the-art mobile CNNs. PPP-Net has also adopted the progressive search strategy used in a recent literature (Liu et al. (2017a)). Experimental results demonstrate that PPP-Net achieves better performances in both (a) higher accuracy and (b) shorter inference time, comparing to the state-of-the-art CondenseNet.

## 1 Introduction

While designing the architectures of neural networks (NNs) has been treated more like an art, the emergence of more complex, sophisticated architectures has posed increasingly bigger challenges for deep-learning practitioners, especially when platform-related constraints (*e.g.*, latency) are in presence. To overcome these challenges, new operations Howard et al. (2017); Zhang et al. (2017); Huang et al. (2017a) have been designed to achieve higher computing efficiency than conventional convolution. Designing these operations requires both profound domain knowledge and intensive human efforts. Therefore, how to automatically generate a neural network—(a) achieves state-of-the-art accuracy and (b) conforms to platform-related constraints—remains as an open & challenging question.

Recently, neural architecture search (NAS) has been proposed to generate network architectures that achieve (or even beat) state-of-the-arts crafted by exports. Works in the field are usually divided into two categories: Reinforcement Learning (RL) based approaches (see *e.g.*, Zoph & Le (2016); Baker et al. (2016); Zoph et al. (2017); Zhong et al. (2017)) and Genetic Algorithm (GA) based approaches (see *e.g.*, Real et al. (2017); Xie & Yuille (2017); Liu et al. (2017b); Real et al. (2018)). One exception is the work by Liu et al. (2017a), which achieves comparable performance to the state-of-the-art RL-based method by using a much more efficient search algorithm. However, all the aforementioned works focus on optimizing only one single objective (*e.g.*, accuracy). There is also one previous work Kim et al. (2017) that searches network architectures by considering both accuracy and inference time. Nevertheless, the training computational power required by their algorithm is very significant, whereas their search space is naively small.

We propose Platform-aware Progressive search for Pareto-optimal Net (PPP-Net) – an efficient algorithm to search Pareto-optimal architectures under multiple objectives (*e.g.*, inference time and accuracy). We define our search space by taking inspirations from state-of-the-art mobile CNNs, which is more compact and efficient compares to usual NAS architectures. We have also adopted the progressive search strategy used in Liu et al. (2017a) to speed up the search process. Experimental results show that our method is able to discover architectures with better accuracy and faster inference time comparing to the baseline approaches.

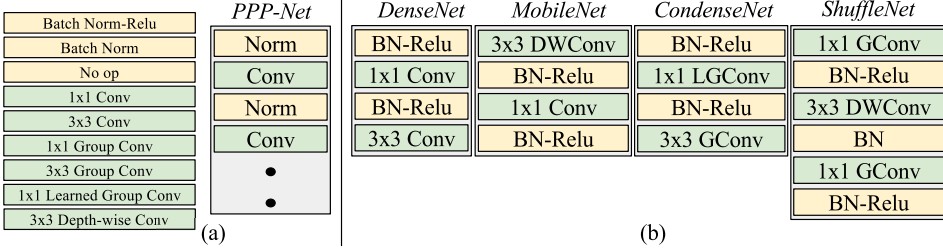

(a)                                              (b)

Figure 1:   Panel (a): Search space design. We show the available layer operations of normalization (yellow boxes) and convolutional (green boxes) (Left) and the block structure of PPP-Net (Right). Panel (b): blocks of efficient CNNs. BN, DW, LG, G stands for Batch Norm, Depth-wise, Learned Group, Group, respectively. All the group convolutions are implicitly followed by channel shuffle operation.

## 2   APPROACH

We propose Platform-aware Progressive search for Pareto-optimal Net (PPP-Net) – a framework automatically generates neural networks with a predefined number of replicated blocks. PPP-Net searches for block architectures to achieve Pareto-optimal performance over multiple objectives.

**Search space design.** Each block consists of multiple layers of two types - normalization (Norm) and convolutional (Conv) layers. We progressively add layers following the Norm-Conv-Norm-Conv order (Fig.1(a)-Right). The operations available for Norm (yellow boxes) and Conv (green boxes) layers are shown in Fig.1(a)-Left. The block of other efficient CNNs are shown in Fig.1(b). Our search space covers hand-crafted efficient operations to take advantages of prior human knowledge on designing efficient CNNs. This not only ensures good quality of our searched architectures but also reduces the searching time for PPP-Net.

**Search Algorithm.** Inspired by Liu et al. (2017a), we adopt Sequential Model-Based Optimization (Hutter et al. (2011)) algorithm to search efficiently with the following four steps (Fig. 2(a)).

1. *Mutate*. For each $\ell$-layers block, we enumerate all possible $\ell + 1$-layers blocks. Before mutation, we keep $K$ models. After mutation, we have $K'$ models where

$$K' = \begin{cases} K \times |Norm|, & \text{if } \ell \bmod 2 = 0 \\ K \times |Conv|, & \text{otherwise} \end{cases} \tag{1}$$

2. *Regress accuracy*. We use a Recurrent Neural Network (RNN) to regress network accuracy given its architecture. This avoids time-consuming training to obtain true accuracy of a network with a slight drawback of regression error.

3. *Select networks*. Our main contribution is to use Pareto Optimality over multiple objectives to select $K$ networks (Fig.. 2(b)) rather than simply select top $K$ accurate ones as in Liu et al. (2017a). Note that other objectives like the number of parameters, FLOPs, and actual inference time can be computed very efficiently.

4. *Update regressor*. We train the selected $K$ networks each for $N$ epochs. Then, we use the evaluation accuracies (output) and the architectures (inputs) to update the RNN regressor.

Since we do not have a well-trained RNN at the beginning, we enumerate all possible 3-layers blocks ($|Norm| \times |Conv| \times |Norm|$ blocks) and train them for $N$ epochs. Then, we use the evaluation

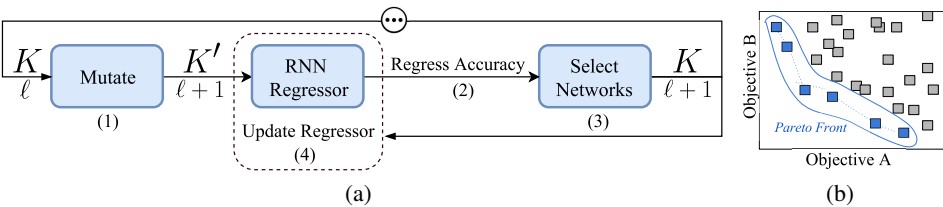

Figure 2: **Search algorithm**. (a) is the flow diagram, $\ell$ is the layers in a block, $K$ is the number of models to train, and $K'$ is the number of models after *Mutate*. (b) is a symbolic figure of Pareto Optimality with two objectives.

accuracies (output) and the architectures (inputs) to train the initial RNN regressor. Next, we follow the four steps described above.

## 3 EXPERIMENTS

We conduct experiments on the CIFAR-10 dataset with standard augmentation. We aim at finding different models that have comparably high accuracy but each possesses a unique characteristic (*e.g.*, the number of parameters is fewer). In the experiment, we end searching at $\ell = 4$, number of epochs $N$ is set to 10, number of models to train, $K$, is set to 128. For selecting the models, we consider evaluation error rate, number of parameters, FLOPs, and actual inference time on our computing platform [1] as the objectives. The output of our Pareto Optimality selection is plotted in Fig.3, we further train the models on the Pareto front for 300 epochs and report the final results in Table.1

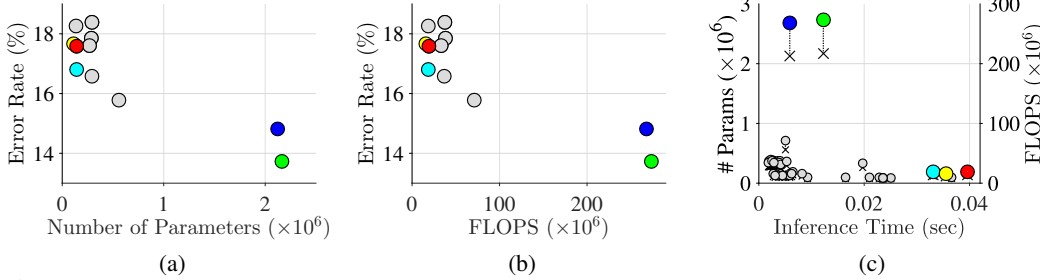

Figure 3: **Pareto front visualization**. (a), (b) is evaluation error rate v.s. parameters, FLOPs, respectively. (c) is parameters and FLOPs v.s. actual inference time, where the dot is params v.s. inference time and the cross is FLOPs v.s. inference time of the same model. The green, yellow, cyan, and blue dots are the PPP-Net-Baseline, PPP-Net-A, PPP-Net-B, and PPP-Net-C, respectively. Finally, CondenseNet (red dot) is included for comparison.

In Fig.3(a,b), it is clear that error rate is inversely proportional to the number of parameters and FLOPs (i.e., the larger the FLOPs, the lower the error rate). However, inference time is not simply proportional to other platform-agnostic objectives such as FLOPs and number of parameters (Fig.3(c)) [2]. Our results demonstrate that inference time is platform-aware since it depends on the software and hardware implementation of the computing platform. For a better comparison, we also plot the CondenseNet (also in our search space) performance in the plot even though it is not in the Pareto front. In Table.1, our results (last group) are compared with state-of-the-art hand-crafted mobile CNNs (middle group) and models using architecture search methods (first group). The inference time of CondenseNet is measured from their official open-sourced code run on our platform. As for our searched models, PPP-Net-Baseline is conducted by choosing the max predicted accuracy architecture without using Pareto Optimality selection. PPP-Net-A, PPP-Net-B, and PPP-Net-C are chosen from the Pareto front where PPP-Net-A has 0.86x parameters and FLOPs of CondenseNet-86 with a comparable performance, PPP-Net-B has 0.65x parameters and FLOPs of CondenseNet-110 with a better performance, and PPP-Net-C is the best accuracy we can find in our Pareto front. Our PPP-Net clearly strikes better trade-off among multiple objectives.

Table 1: **Multiple Objectives Comparison**. Missing values are the metrics not reported in their original papers. Pareto front visualizations of our searched networks can also be found in Fig. 3.

| Model | Error rate | Params | FLOPs | Time |
|---|---|---|---|---|
| Real et al. Real et al. (2017) | 5.4 | 5.4M | - | - |
| NASNet-B Zoph et al. (2017) | *3.73* | 2.6M | - | - |
| PNASNet-1 Liu et al. (2017a) | *4.01* | 1.6M | - | - |
| DenseNet (k=12) Huang et al. (2017b) | 5.24 | 1.0M | - | - |
| CondenseNet-110 Huang et al. (2017a) | 4.63 | 0.79M | 98.2M | 0.040 |
| CondenseNet-98 Huang et al. (2017a) | 4.83 | 0.65M | 81.3M | 0.033 |
| CondenseNet-86 Huang et al. (2017a) | 5.0 | 0.52M | 65.8M | 0.028 |
| PPP-Net-Baseline | **4.36** | 11.39M | 1364M | 0.035 |
| PPP-Net-A | 5.28 | **0.45M** | **56.9M** | **0.025** |
| PPP-Net-B | 4.58 | 0.52M | 63.5M | 0.026 |
| PPP-Net-C | **4.36** | 11.29M | 953M | 0.030 |

---

[1] We run our training and inference on NVIDIA Titan X Pascal GPU.

[2] Our search algorithm is in Tensorflow and Table.1 is in PyTorch, which results in different inference time.

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
