# OpenReview forum: "PPP-Net: Platform-aware Progressive Search for Pareto-optimal Neural Architectures"
_ICLR.cc/2018/Workshop — Accept_

### Official Review · AnonReviewer1 · 2018-03-09
**This paper targets on a very important practical problem which is platform-aware neural architecture search. The idea is rather straightforward. And experiments are conducted on CIFAR-10 dataset under the framework of TensorFlow. However, the results are not promising enough.**

**Rating:** 5
**Confidence:** 4

**Review:**

This paper targets on a very important practical problem which is platform-aware neural architecture search.

The motivation is clear and the writing is easy to understand.

If we consider inference time as an objective function that is similar to GFLOPS or accuracy. Then there will be no much difference between them. However, in real applications, even the network architecture is given, there are still many other factors that will affect the inference time, such as different platforms (GPU/CPU/ARM/ASIC),  different frameworks or optimized tools,  different input scales, different batch sizes and so on.

For the network architecture, only sequential networks are considered, and only toy experiments on Cifar-10 are conducted.

Overall, there are more work should be done to demonstrate the effectiveness the proposed idea.

---

### Official Review · AnonReviewer2 · 2018-03-09
**An extension for efficient neural architecture search for CNN using the pareto optimality.**

**Rating:** 7
**Confidence:** 2

**Review:**

The paper presents an extension of the method in Liu et at (2017a) by using a Pareto optimal search under multiple objectives fron efficient neural architecture search for CNNs. The paper is well written and clear, towards its objections and added value. Furthermore, the authors evaluated their proposed method using the CIFAR-10 dataset with standard augmentation and compared with both SOTA hand-crafted mobile CNNs, as well models using architecture search methods. The proposed method is really interesting and provides a useful approach how to automatically generate a neural network that achieves SOTA accuracy but also conforms to other platform-related constrains.

---

### Official Review · AnonReviewer3 · 2018-03-09
**This submission proposes a technique to search optimal neural architectural, and simultaneously, taking platform-related constrictions into considerations.**

**Rating:** 6
**Confidence:** 1

**Review:**

This submission proposes a technique to search optimal neural architecture, and simultaneously, take platform-related constrictions into considerations.

-- Pros:
    This paper tries to propose a technique to automatically find an optimal neural architecture. Based on the experimental results, their method seems effective.

I am not familiar with this research direction so I can only make an educated guess.

---

### Decision · Program_Chairs · 2018-03-20
**ICLR 2018 Workshop Acceptance Decision**

**Decision:**

Accept

**Comment:**

Congratulations, your paper was accepted to the ICLR workshop.